# Evidence for absence of links between striatal dopamine synthesis capacity and working memory capacity, spontaneous eye-blink rate, and trait impulsivity

Ruben van den Bosch[1]*, Frank H Hezemans[1], Jessica I Määttä[2], Lieke Hofmans[3], Danae Papadopetraki[1,4], Robbert-Jan Verkes[4], Andre F Marquand[1], Jan Booij[5,6], Roshan Cools[1,4]

[1]Donders Institute for Brain, Cognition and Behaviour, Radboud University Nijmegen, Nijmegen, Netherlands; [2]Department of Psychology, Stockholm University, Stockholm, Sweden; [3]Department of Developmental Psychology, University of Amsterdam, Amsterdam, Netherlands; [4]Department of Psychiatry, Radboud University Nijmegen Medical Centre, Nijmegen, Netherlands; [5]Department of Radiology and Nuclear Medicine, Amsterdam University Medical Centers, Amsterdam, Netherlands; [6]Department of Medical Imaging, Radboud University Medical Center, Nijmegen, Netherlands

*For correspondence:
ruben.vandenbosch@donders.
ru.nl

Competing interest: The authors declare that no competing interests exist.

**Abstract** Individual differences in striatal dopamine synthesis capacity have been associated with working memory capacity, trait impulsivity, and spontaneous eye-blink rate (sEBR), as measured with readily available and easily administered, 'off-the-shelf' tests. Such findings have raised the suggestion that individual variation in dopamine synthesis capacity, estimated with expensive and invasive brain positron emission tomography (PET) scans, can be approximated with simple, more pragmatic tests. However, direct evidence for the relationship between these simple trait measures and striatal dopamine synthesis capacity has been limited and inconclusive. We measured striatal dopamine synthesis capacity using [18F]-FDOPA PET in a large sample of healthy volunteers (N = 94) and assessed the correlation with simple, short tests of working memory capacity, trait impulsivity, and sEBR. We additionally explored the relationship with an index of subjective reward sensitivity. None of these trait measures correlated significantly with striatal dopamine synthesis capacity, nor did they have out-of-sample predictive power. Bayes factor analyses indicated the evidence was in favour of absence of correlations for all but subjective reward sensitivity. These results warrant caution for using these off-the-shelf trait measures as proxies of striatal dopamine synthesis capacity.

## Editor's evaluation

This study presents fundamental insights into the relationship between [18F]-FDOPA PET measurements of striatal dopamine synthesis capacity and a series of measures, including behavioral readouts, proposed to index dopamine function. The frequentist and Bayesian analyses together provide compelling evidence for an absence of any relationship between striatal dopamine synthesis capacity and external measures, questioning the interpretation of studies using such measures to index dopamine function. These findings will not only be of great interest to cognitive neuroscientists but also inform future studies of neuropsychiatric diseases.

## Introduction

The mesocorticolimbic dopamine system plays a key role in a range of cognitive functions, including cognitive control processes, such as working memory (*Arnsten and Li, 2005*), attention (*Thiele and Bellgrove, 2018*), and flexible behaviour (*Floresco and Magyar, 2006*). While such processes have classically been associated with prefrontal dopamine function, they also critically depend on the basal ganglia and dopamine activity in the striatum (*Cools, 2019*; *Frank and O'Reilly, 2006*; *Ott and Nieder, 2019*). Accordingly, individual differences in striatal dopamine function have been associated with various behavioural and physiological trait characteristics, including working memory capacity, impulsivity, and spontaneous eye-blink rate (sEBR). We focused on striatal dopamine and investigated the relationships suggested by prior literature between these trait characteristics and individual variation in striatal dopamine function.

For example, evidence from work with brain positron emission tomography (PET) in humans indicates that working memory capacity is positively associated with both striatal dopamine synthesis capacity, assessed with 6-[$^{18}$F]fluoro-l-m-tyrosine ([$^{18}$F]-FMT) PET imaging, and striatal dopamine release, estimated with [$^{11}$C]-raclopride PET imaging (*Bäckman et al., 2011*; *Clatworthy et al., 2009*; *Cools et al., 2008*; *Landau et al., 2009*; *Salami et al., 2019*; but see *Braskie et al., 2008*, *Braskie et al., 2011*; *Klostermann et al., 2012*). These studies observed that greater dopamine synthesis or release, typically in the caudate nucleus, was associated with improved updating of working memory in particular (*Bäckman et al., 2011*), which is in line with computational modelling and neuroimaging work indicating the relevance of the basal ganglia for working memory updating (*Frank and O'Reilly, 2006*; *McNab and Klingberg, 2008*; *Nir-Cohen et al., 2020*). The PET findings are also corroborated by studies with unmedicated patients with Parkinson's disease that show striatal underactivation and impaired performance during working memory updating (*Marklund et al., 2009*), as well as from experimental animal work demonstrating that lesions of the caudate nucleus cause impairment of working memory performance (*Collins et al., 2000*). Furthermore, genetic variation in striatal dopamine function has been linked to working memory performance (*Frank and Fossella, 2011*) and associated striatal (caudate) activation in working memory updating tasks (e.g. *Stollstorff et al., 2010*).

Work with PET imaging in human volunteers has also demonstrated a positive correlation between impulsivity and both striatal dopamine release and synthesis capacity as well as striatal dopamine D$_2$-receptor availability (*Buckholtz et al., 2010*; *Kim et al., 2014*; *van Holst et al., 2018*). This is supported by evidence from a meta-analysis showing that patients with Parkinson's disease who developed treatment-induced impulse control disorders had reduced striatal dopamine transporter availability and increased dopamine release in the ventral striatum in response to reward-related stimuli (*Martini et al., 2018*). Such findings provide a neural mechanistic account of observed associations between working memory and impulsivity (*Cools et al., 2007*; *James et al., 2007*), as well as working memory and impulse control deficits in disorders that implicate abnormal striatal dopamine signalling, such as schizophrenia (*Barch and Ceaser, 2012*), drug addiction (*Jentsch and Taylor, 1999*) and Parkinson's disease (*Cools et al., 2022*).

Intriguingly, some of the cognitive factors associated with striatal dopamine transmission have been shown to be captured by sEBR. Individual variation in sEBR has been found to correlate with working memory task performance (*Ortega et al., 2022*), working memory updating and gating (*Rac-Lubashevsky et al., 2017*), attentional load and fatigue (*Maffei and Angrilli, 2018*), attentional control (*Colzato et al., 2009*; *Unsworth et al., 2019*), and exploration during reinforcement learning (*Van Slooten et al., 2019*). Furthermore, variability in dopaminergic drug effects on performance on these types of tasks can be partly explained by the individual variation in sEBR (e.g. *Cavanagh et al., 2014*). Given such findings, it has been argued that sEBR is an effective measure of striatal dopamine activity (*Jongkees and Colzato, 2016*), although the exact mechanism by which they are connected remains unclear. The notion is supported, however, by positive correlations between sEBR and dopamine D$_2$-receptor availability throughout the striatum (*Groman et al., 2014*; but see *Dang et al., 2017*), as well as post-mortem dopamine concentrations in the caudate nucleus of monkeys (*Taylor et al., 1999*).

Findings like these have raised the suggestion that instead of requiring invasive and expensive brain PET imaging, we can approximate striatal dopamine activity with relatively simple, off-the-shelf tests, for example, the digit span or listening span tests of working memory capacity (*Cools et al., 2008*; *Landau et al., 2009*), eye-blink rate measurements at rest (*Jongkees and Colzato, 2016*),

or self-report measures of relevant traits such as impulsivity or reward sensitivity (*Buckholtz et al., 2010*; *Clatworthy et al., 2009*; *Depue and Collins, 1999*; *Froböse et al., 2018*). If this were true, then this would have great implications for a variety of disciplines, including clinical and cognitive psychology, neuroscience, and experimental medicine. After all, it would mean that such inexpensive and noninvasive tests could be used as proxy measures for studying individual differences in striatal dopamine function and for stratifying dopaminergic drug effects by baseline dopamine levels. Indeed, we and others have interpreted a variety of working memory span-, sEBR-, and impulsivity-dependent changes in behaviour (e.g. elicited by drug administration) as reflecting individual variation in baseline striatal dopamine function (*Cavanagh et al., 2014*; *Clatworthy et al., 2009*; *Colzato et al., 2008*; *Cools et al., 2007*; *Frank and O'Reilly, 2006*; *Froböse et al., 2018*; *Rostami Kandroodi et al., 2021*; *Kimberg et al., 1997*; *Slagter et al., 2015*; *Swart et al., 2017*; *van der Schaaf et al., 2014*).

However, the predictive value of working memory capacity, trait impulsivity, and sEBR for individual variation in striatal dopamine function in the human brain has not yet been established. This requires assessing the relationships of these measures with striatal dopamine function measured with PET imaging in studies with much larger sample sizes than used so far, as well as testing the predictive accuracy of the putative proxy measures for previously unseen data (*Yarkoni and Westfall, 2017*). In fact, sample sizes of prior studies of these measures have been so small that they might well have produced inflated and unstable estimates of the correlation coefficient (*Loken and Gelman, 2017*; *Schönbrodt and Perugini, 2013*). For example, although both *Cools et al., 2008* and *Landau et al., 2009* (N = 11 and N = 22, respectively) reported a positive correlation between working memory capacity, measured with the listening span, and dopamine synthesis capacity, subsequent studies using a partly overlapping sample did not find this relationship (*Braskie et al., 2011*; *Braskie et al., 2008*; *Klostermann et al., 2012*; all using subsets from one sample of N = 37 participants, of which 20 overlapped with *Landau et al., 2009*). Similarly, trait impulsivity has also been observed to correlate negatively rather than positively with striatal dopamine levels, synthesis capacity, and $D_2$-receptor availability (*Dalley et al., 2007*; *Martinez et al., 2020*; *Petzold et al., 2019*, N = 60; *Smith et al., 2016*, N = 16). Furthermore, *Sescousse et al., 2018* reported evidence for the absence of a correlation between sEBR and striatal dopamine synthesis capacity (N = 30), similar to the lack of a correlation between sEBR and dopamine $D_2$-receptor availability (N = 20) observed by *Dang et al., 2017*.

Here, we measured striatal dopamine synthesis capacity with [$^{18}$F]-FDOPA PET imaging in a much larger sample of healthy volunteers (N = 94) to establish the predictive link with three commonly used putative proxy measures of striatal dopamine: working memory capacity, trait impulsivity, and sEBR. In addition, we explored the potential correlation between striatal dopamine synthesis capacity and a questionnaire index of subjective reward sensitivity, a process that is intimately linked with dopamine function (*Costumero et al., 2013*; *Depue and Collins, 1999*; *Locke and Braver, 2008*). The subcortical dopamine system is not a single entity and the striatum is a functionally heterogeneous structure with a distinct connectionist anatomy with the cortex, involving a functional gradient in the connections between the cortex and ventral striatum (primarily nucleus accumbens), dorsolateral striatum (putamen), and dorsomedial striatum (caudate nucleus; *Alexander et al., 1986*; *Haber et al., 2000*; *Joel and Weiner, 2000*). To take this heterogeneity into account, we performed our analyses in three striatal regions of interest (ROIs), defined using a parcellation based on intra-striatal functional connectivity in an independent sample (*Piray et al., 2017*). The ROIs approximately matched the anatomical subdivision of the striatum into caudate nucleus, putamen, and nucleus accumbens (ventral striatum). We tested the hypotheses that the trait measures were positively correlated with, and predictive of, estimates of striatal dopamine synthesis capacity in these striatal ROIs.

## Results

### Absence of correlations between trait measures and striatal dopamine synthesis capacity

First, we report the Pearson correlations between striatal dopamine synthesis capacity and working memory capacity, trait impulsivity, sEBR, and subjective reward sensitivity. Dopamine synthesis capacity was quantified as the [$^{18}$F]-FDOPA influx rate ($k_i^{cer}$) for our three striatal ROIs (*Piray et al., 2017*): the caudate nucleus, putamen, and nucleus accumbens (ventral striatum). Working memory capacity was indexed with the Digit Span test (*Groth-Marnat, 1997*) and Listening Span test

(*Daneman and Carpenter, 1980*); trait impulsivity was assessed with the Barratt Impulsiveness Scale (BIS-11; *Patton et al., 1995*); subjective reward sensitivity was assessed using the Behavioural Activation Scale (BAS; *Carver and White, 1994*); and sEBR was calculated from electro-oculography recordings (see 'Methods' for details). For trait measures that resulted in multiple (sub)scores, we used the total or composite score as the putative proxy measure. We report additional correlation analyses exploring the relationships between subscales of the trait measures and dopamine synthesis capacity in *Figure 1—figure supplements 1–3*.

There were no significant correlations between dopamine synthesis capacity, in any of the three striatal ROIs, and working memory capacity, trait impulsivity, or sEBR, even without correcting for multiple comparisons (*Figure 1*, *Table 1*; all uncorrected p-values>0.050, all corrected p-values≥0.365). There was also no significant correlation with subjective reward sensitivity after correction for multiple comparisons (*Figure 1*, *Table 1*; corrected p-values=0.127). Specifically, the coefficients of correlation between dopamine synthesis capacity and both the Digit Span and Listening Span were almost zero in all three ROIs (*Figure 1a and b*, *Table 1*). In fact, the direction of the association between Digit Span and dopamine synthesis capacity was negative (e.g. $\rho = -0.071$ for the nucleus accumbens). The coefficient of the correlation between trait impulsivity and dopamine synthesis capacity was also almost zero in all three ROIs, and the direction of the association was, if anything, negative (*Figure 1c*, *Table 1*). There were also no significant correlations between sEBR and dopamine synthesis capacity (*Figure 1d*, *Table 1*), with correlation coefficients of almost zero in the caudate nucleus and putamen, and a positive (but non-significant) association in the nucleus accumbens ($\rho = 0.123$). That association was slightly stronger but still not significant after correction for multiple comparisons when excluding nine participants that received inconsistent instructions for the sEBR measurement (see 'Methods'; $\rho = 0.184$, adjusted p-value=0.143). Among the four trait measures under consideration, subjective reward sensitivity was most strongly and positively associated with dopamine synthesis capacity (e.g. $\rho = 0.179$ in the nucleus accumbens), but again the correlations were not statistically significant (*Figure 1e*, *Table 1*).

In accordance with the ROI-based correlation analyses, voxel-wise regression analyses of the trait measures on the dopamine synthesis capacity PET data did not reveal any significant clusters, and the location of any sub-threshold clusters was generally consistent with the ROI-based results (*Figure 2*).

Because of the lack of statistically significant correlations, we sought to quantify the evidence in our data for the absence of correlations between the trait measures and dopamine synthesis capacity using a Bayesian analysis of the associations. We calculated the Bayes factors ($BF_{01}$) of the null hypotheses that dopamine synthesis capacity values and the different trait measures are not positively correlated (H0) versus the alternative hypotheses that they are positively correlated (H1; *Table 2*). For working memory capacity, the $BF_{01}$ results indicated that the data were approximately 10–12 times more likely under the null hypothesis of no positive correlation than under the alternative hypothesis of a positive correlation for the Digit Span, and approximately 6 times more likely under the null hypothesis for the Listening Span, thus providing strong evidence for H0 over H1 for the Digit Span and moderate evidence for the Listening Span. Similarly, there was strong evidence for the null hypothesis of no positive correlation between trait impulsivity and dopamine synthesis capacity ($BF_{01} \approx 5$–8). For sEBR, the analysis revealed moderate evidence for the null hypothesis of no positive correlation with dopamine synthesis capacity in the caudate nucleus and putamen ($BF_{01} \approx 6$), but only weak, inconclusive evidence in the nucleus accumbens ($BF_{01} \approx 2$). The data for the correlation between subjective reward sensitivity and dopamine synthesis capacity were equally likely under H0 or H1 ($BF_{01} \approx 1$), indicating that the Bayesian analysis was inconclusive about the evidence for the existence of a positive correlation.

To assess the sensitivity of the Bayes factors to the priors that were used in the analyses, we performed a Bayes factor robustness check. This analysis indicated that the above conclusions hold even when specifying strong prior beliefs in the existence of positive correlations (*Figure 1—figure supplement 4*).

We also assessed Bayes factors quantifying the relative evidence for the non-directional hypotheses of no correlation (H0) versus a (positive or negative) correlation (H1) between the trait measures and dopamine synthesis capacity (*Table 2*). This revealed moderate evidence for the null hypothesis of no correlation (H0) between striatal dopamine synthesis capacity and working memory capacity ($BF_{01}$

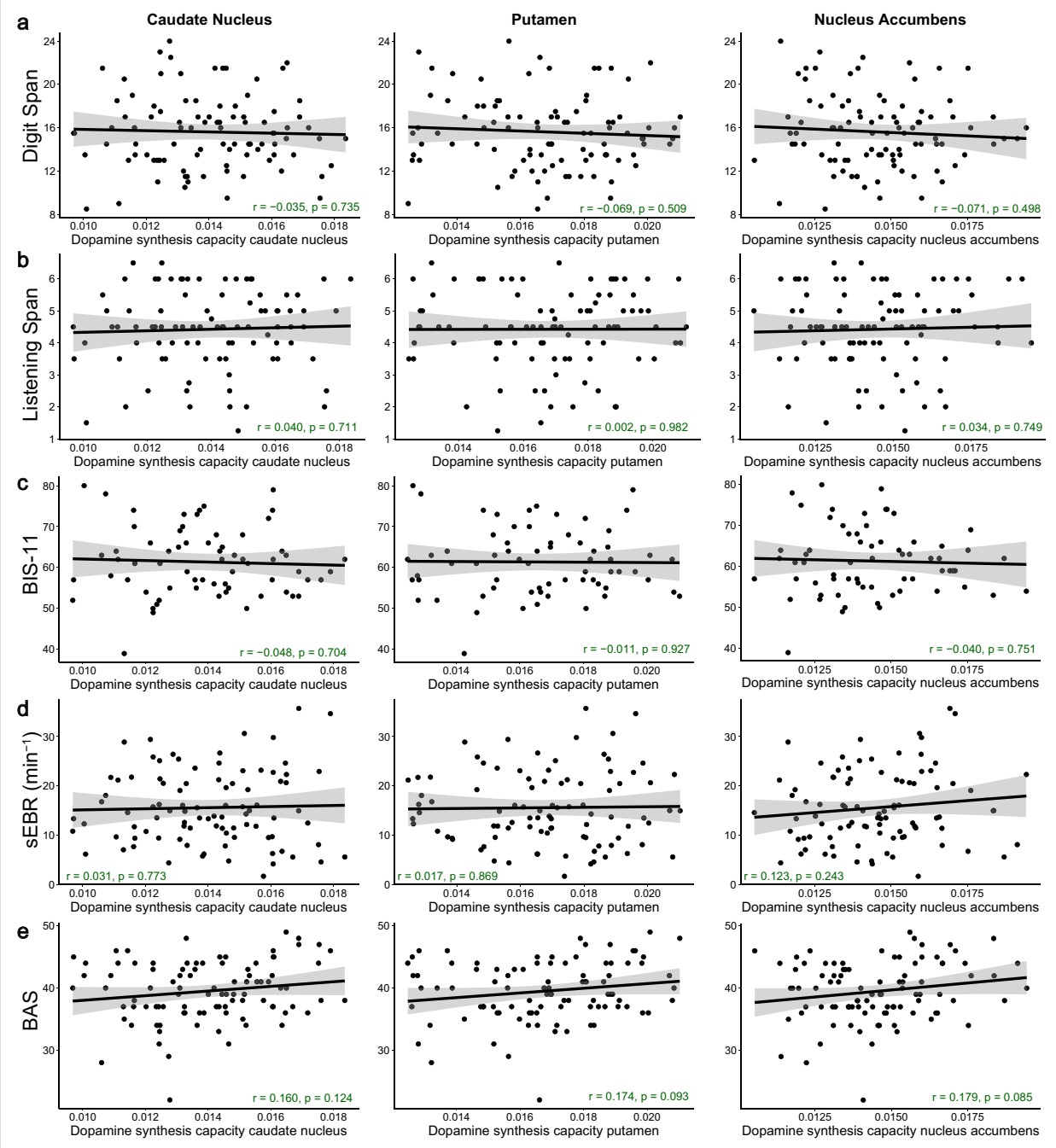

**Figure 1.** No significant correlations between striatal dopamine synthesis capacity and working memory capacity, trait impulsivity, spontaneous eye-blink rate, or subjective reward sensitivity. Pearson correlations between dopamine synthesis capacity ($k_i^{cer}$) in the caudate nucleus, putamen, or nucleus accumbens regions of interest (ROIs) and (**a**) working memory capacity measured with the Digit Span task (N = 94), (**b**) working memory capacity measured with the Listening Span task (N = 94), (**c**) trait impulsivity measured with the BIS-11 questionnaire (N = 66), (**d**) spontaneous eye-blink rate (N = 92), or (**e**) subjective reward sensitivity measured with the Behavioural Activation Scale (N = 94). The light grey shading represents the 95% confidence interval. The p-values provided in the annotations are not corrected for multiple comparisons.

The online version of this article includes the following figure supplement(s) for figure 1:

**Figure supplement 1.** No significant correlations between striatal dopamine synthesis capacity and subsections of the working memory tasks (N = 94).

**Figure supplement 2.** No significant correlations between striatal dopamine synthesis capacity and subsections of the BIS-11 questionnaire of trait impulsivity (N = 66).

*Figure 1 continued on next page*

*Figure 1 continued*

**Figure supplement 3.** No significant correlations between striatal dopamine synthesis capacity and subsections of the BAS questionnaire of trait reward sensitivity (N = 94).

**Figure supplement 4.** Bayes factor robustness checks.

**Figure supplement 5.** Sampling variability of the correlation between dopamine synthesis and trait measures.

**Figure supplement 6.** Distribution of dopamine synthesis capacity ($k_i^{cer}$) values in the striatal regions of interest (ROI): caudate nucleus, putamen, and nucleus accumbens.

**Figure supplement 7.** Analysis masks of the striatal regions of interest: caudate nucleus (red), putamen (green), and nucleus accumbens (ventral striatum; blue).

≈ 5–7), trait impulsivity ($BF_{01} \approx 6$), or sEBR ($BF_{01} \approx 4$–8). For subjective reward sensitivity, the relative evidence for the null hypothesis remained inconclusive ($BF_{01} \approx 2$).

Of note, previous studies that used smaller sample sizes typically reported much larger correlation coefficients than those observed in this study (*Figure 1*, *Table 1*; range: –0.071–0.179). One potential explanation for this apparent contradiction is sampling variability: the estimates of effect size can be inflated due to chance with small samples and increasingly large samples are needed to sufficiently reduce estimated standard errors to detect smaller effects (*Marek et al., 2022*; *Schönbrodt and Perugini, 2013*). To illustrate this issue, we repeatedly sampled (with replacement) random subsets of varying sizes from our full dataset. For each of those random subsets, we estimated the correlation coefficients between each trait measure and striatal dopamine synthesis capacity, so that we could examine the sampling variability of the correlation coefficient as a function of sample size. As expected, this analysis revealed substantial sampling variability at smaller sample sizes, such that two

**Table 1.** Statistics of the Pearson correlation analyses between striatal dopamine synthesis capacity and working memory capacity, trait impulsivity, spontaneous eye-blink rate, and subjective reward sensitivity.

| Trait measure | Striatal ROI | ρ | p-value | Adjusted p-value | $BF_{01}$ (0,1) | $BF_{01}$ (–1,1) |
|---|---|---|---|---|---|---|
| Digit Span | Caudate nucleus | –0.035 | 0.632 | 1 | 9.93 | 7.33 |
| | Putamen | –0.069 | 0.746 | 1 | 12.19 | 6.25 |
| | Nucleus accumbens | –0.071 | 0.751 | 1 | 12.31 | 6.19 |
| Listening Span | Caudate nucleus | 0.04 | 0.355 | 1 | 5.52 | 7.09 |
| | Putamen | 0.002 | 0.491 | 1 | 7.45 | 7.59 |
| | Nucleus accumbens | 0.034 | 0.375 | 1 | 5.78 | 7.22 |
| BIS-11 | Caudate nucleus | –0.048 | 0.648 | 1 | 8.56 | 6.06 |
| | Putamen | –0.011 | 0.536 | 1 | 6.98 | 6.48 |
| | Nucleus accumbens | –0.04 | 0.625 | 1 | 4.97 | 6.19 |
| sEBR | Caudate nucleus | 0.031 | 0.386 | 0.773 | 6.01 | 7.36 |
| | Putamen | 0.017 | 0.434 | 0.773 | 6.7 | 7.57 |
| | nucleus accumbens | 0.123 | 0.122 | 0.365 | 2.24 | 3.92 |
| BAS | caudate nucleus | 0.16 | 0.062 | 0.127 | 1.3 | 2.42 |
| | putamen | 0.174 | 0.046 | 0.127 | 1.02 | 1.93 |
| | nucleus accumbens | 0.179 | 0.042 | 0.127 | 0.94 | 1.8 |

ρ: Pearson correlation coefficient; corresponding one-sided p-values (for a positive association) are reported both with and without Holm–Bonferroni correction for multiple comparisons for three striatal regions of interest (ROIs); $BF_{01}$ (0,1): Bayes factor for the evidence in favour of the null hypothesis of no positive correlation (H0) versus the alternative hypothesis of a positive correlation (H1); $BF_{01}$ (–1,1): Bayes factor for the non-directional hypotheses quantifying the evidence in favour of the null hypothesis of no correlation (H0) versus the alternative hypothesis that there is a correlation (H1); BIS-11: Barratt Impulsiveness Scale, assessing trait impulsivity; sEBR: spontaneous eye-blink rate; BAS: Behavioural Activation Scale, assessing subjective reward sensitivity.

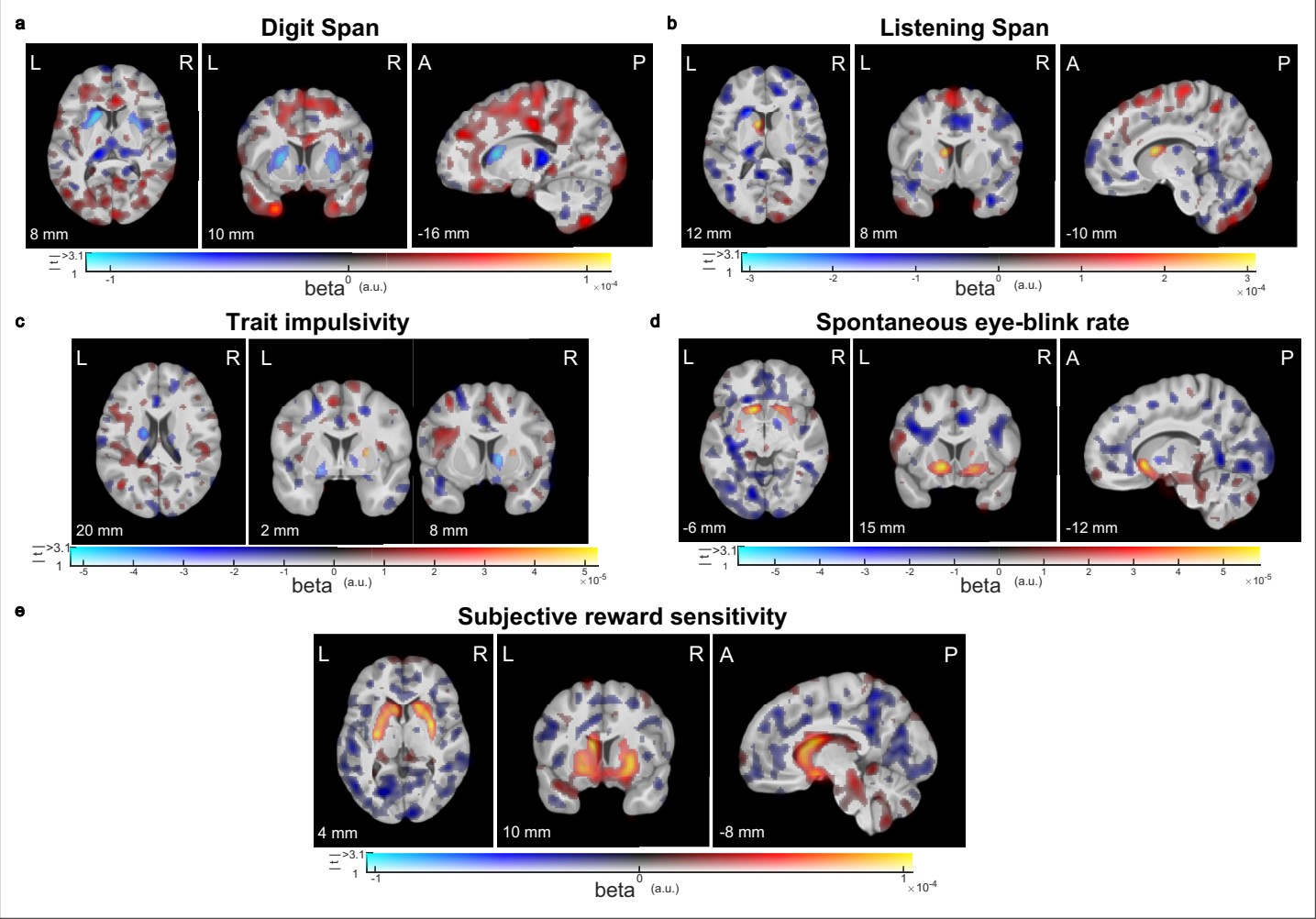

**Figure 2.** Visualization of sub-threshold correlations between striatal dopamine synthesis capacity and working memory capacity, trait impulsivity, spontaneous eye-blink rate, or subjective reward sensitivity. Results from voxel-wise regression analysis of the trait measures onto the PET index of dopamine synthesis capacity for (**a**) working memory capacity measured with the Digit Span task (N = 94), (**b**) working memory capacity measured with the Listening Span task (N = 94), (**c**) trait impulsivity measured with the BIS-11 questionnaire (N = 66), (**d**) spontaneous eye-blink rate (N = 92), or (**e**) subjective reward sensitivity measured with the Behavioural Activation Scale (N = 94). In these dual-coded images, colour indicates the size of the contrast estimate and the opacity represents the t-values. Voxels with t-values above the threshold of p<0.001, uncorrected, are fully opaque. There are no significant clusters in these images (peak-level family-wise error correction at p<0.05 after small-volume correction for the combination of the caudate nucleus, putamen, and nucleus accumbens). Any significant clusters would have been encircled in black for red blobs or white for blue blobs. The results are overlaid on the group-average T1-weighted anatomical MRI scan in MNI152 coordinate space. This visualization approach was introduced by *Allen et al., 2012* and implemented by *Zandbelt, 2017*.

independent population subsamples could produce opposite conclusions about the same correlation. For example, at N = 15, the 95% confidence interval for the correlation between sEBR and dopamine synthesis capacity in the nucleus accumbens was [–0.42, 0.62], highlighting how such correlations can be strongly influenced by chance. *Figure 1—figure supplement 5* illustrates the sampling variability by sample size for all trait measures and striatal ROIs.

To further inspect the impact of sample size and power, we calculated the effect size that we would be able to reliably detect with the one-sided Pearson correlation tests at acceptable levels of statistical power in our sample of N = 94 (with statistical significance level of $\alpha$ = 0.05), using G*Power (*Faul et al., 2009*). With power of 0.9 we would be able to reliably detect correlations with a coefficient of $\rho$ = 0.29 (which corresponds to a sample Cohen's d of 0.61). For a power of 0.8, the coefficients would have to be $\rho$ = 0.25 (sample Cohen's d = 0.52). The hypothesized correlations between striatal dopamine synthesis capacity and working memory capacity, impulsivity, and sEBR were all much

**Table 2.** Summary statistics of the cross-validated predictive accuracy of working memory capacity, trait impulsivity, spontaneous eye-blink rate, or subjective reward sensitivity for striatal dopamine synthesis capacity.

| Trait measure | Striatal ROI | Metric: $R^2$ | | | Metric: RMSE | | |
|---|---|---|---|---|---|---|---|
| | | Mean | Std_err | Perm_p | Mean | Std_err | Perm_p |
| Digit Span | Caudate nucleus | –0.19812 | 0.010189 | 0.617 | 0.002054 | 1.21E-05 | 0.6242 |
| | Putamen | –0.19854 | 0.010929 | 0.6048 | 0.002222 | 1.43E-05 | 0.4672 |
| | Nucleus accumbens | –0.20066 | 0.011087 | 0.6154 | 0.001895 | 1.29E-05 | 0.5174 |
| Listening Span | Caudate nucleus | –0.2009 | 0.01004 | 0.6272 | 0.002056 | 1.21E-05 | 0.6686 |
| | putamen | –0.20468 | 0.01074 | 0.6246 | 0.002227 | 1.39E-05 | 0.5344 |
| | nucleus accumbens | –0.20312 | 0.010809 | 0.6262 | 0.001898 | 1.28E-05 | 0.5624 |
| BIS-11 | Caudate nucleus | –0.3554 | 0.019555 | 0.6562 | 0.00212 | 1.46E-05 | 0.71 |
| | Putamen | –0.42078 | 0.032679 | 0.7408 | 0.002298 | 1.60E-05 | 0.7188 |
| | Nucleus accumbens | –0.3709 | 0.02575 | 0.6382 | 0.001977 | 1.50E-05 | 0.6384 |
| sEBR | Caudate nucleus | –0.20731 | 0.010552 | 0.6382 | 0.002077 | 1.22E-05 | 0.748 |
| | Putamen | –0.18209 | 0.010114 | 0.5242 | 0.0022 | 1.31E-05 | 0.5688 |
| | Nucleus accumbens | –0.16276 | 0.008713 | 0.4236 | 0.001874 | 1.31E-05 | 0.335 |
| BAS | Caudate nucleus | –0.16081 | 0.01069 | 0.461 | 0.002022 | 1.27E-05 | 0.187 |
| | Putamen | –0.16465 | 0.011264 | 0.4672 | 0.002189 | 1.44E-05 | 0.1226 |
| | Nucleus accumbens | –0.16523 | 0.011255 | 0.4682 | 0.001865 | 1.29E-05 | 0.1468 |

$R^2$: coefficient of determination; RMSE: root mean square error; std_err: standard error; perm_p: p-value based on 5000 permutations; BIS-11: trait impulsivity questionnaire; sEBR: spontaneous eye-blink rate; BAS: Behavioural Activation Scale to measure subjective reward sensitivity.

weaker than that in the current sample: all but one correlation with $\rho$ < 0.1. For an effect size of $\rho$ = 0.1 and power of 0.8, a sample size of N = 614 would have been needed (N = 850 for power of 0.9).

## Trait measures fail to predict striatal dopamine synthesis capacity

The statistical significance of a correlational model evaluated 'in-sample' does not necessarily speak to that model's predictive accuracy for previously unseen data (*Yarkoni and Westfall, 2017*). We therefore used resampling methods, specifically *k*-fold cross-validation (*k* = 10 with 100 repeats) and permutation testing, to estimate to what extent each of the trait measures could predict striatal dopamine synthesis capacity, using simple linear regression models (see 'Methods' for details). We indexed predictive accuracy with the out-of-sample coefficient of determination ($R^2$) and provide the root mean square error (RMSE) as an additional performance measure (*Table 2*).

For all trait measures, the cross-validated $R^2$ was negative for dopamine synthesis capacity in the caudate nucleus, putamen, and nucleus accumbens ROIs (*Table 2*). This implies that using a trait measure to predict dopamine synthesis capacity resulted in a worse performance than simply predicting the average dopamine synthesis capacity – in other words, ignoring individual differences altogether. The cross-validated RMSE values were approximately 0.002 for all trait measures (*Table 2*), which is a substantial amount of error when considering that the interquartile range of dopamine synthesis capacity was approximately 0.003 (*Figure 1—figure supplement 6*). This is illustrated in *Figure 3*, which shows for each participant the errors of the predictions of dopamine synthesis capacity in the ROIs from each of the trait measures. Good predictive value would have resulted in small variance of the predictions from each repeat around the zero-line for all or nearly all participants. However, the current predictions have large errors and the direction and size of the prediction error vary greatly over participants and trait measures.

Permutation testing indicated that for all trait measures the cross-validated predictive performance was not statistically significant (*Table 2*; all permutation p-values >0.10). That is, the observed

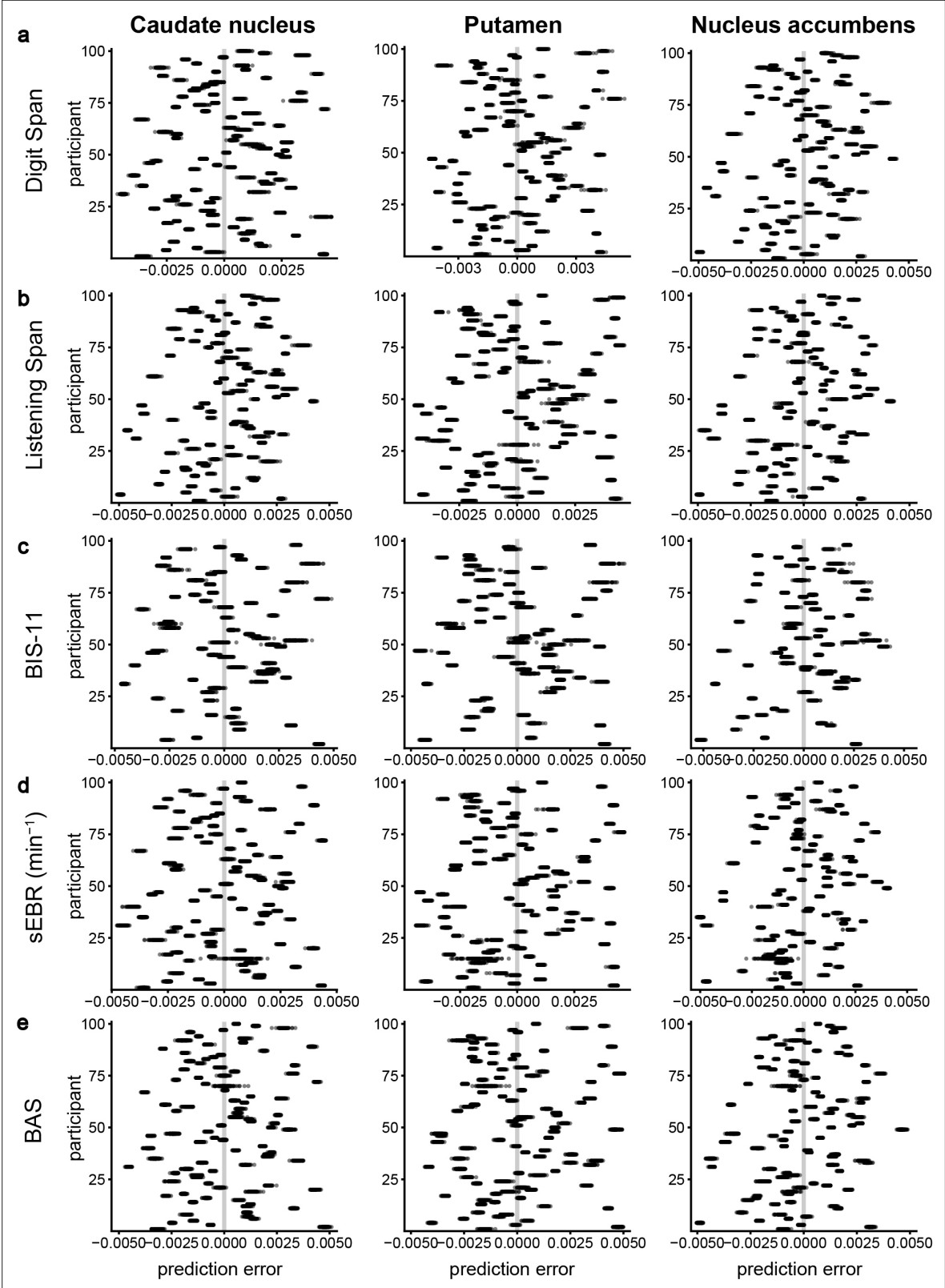

**Figure 3.** Cross-validation results of the predictive performance of working memory capacity, trait impulsivity, spontaneous eye-blink rate, or subjective reward sensitivity for striatal dopamine synthesis capacity. Every plot shows for each participant the error of the prediction of dopamine synthesis capacity in the caudate nucleus (left column), putamen (middle column), or nucleus accumbens regions of interest (ROIs) (right column) from (**a**) working memory capacity measured with the Digit Span task (N = 94), (**b**) working memory capacity measured with the Listening Span task (N = 94), (**c**) trait

*Figure 3 continued on next page*

*Figure 3 continued*

impulsivity measured with the BIS-11 questionnaire (N = 66), (**d**) spontaneous eye-blink rate (N = 92), or (**e**) subjective reward sensitivity measured with the Behavioural Activation Scale (N = 94). Each black dot represents the prediction error of one of 100 repeats of the k-fold cross-validation (k = 10). For each panel, the grey vertical line at zero represents perfect predictive accuracy.

cross-validated $R^2$ and RMSE values were not different from those under the null hypothesis of no predictive power for each of the trait measures.

## Discussion

In this study, we provide evidence for a lack of correlations between striatal dopamine synthesis capacity, measured with [$^{18}$F]-FDOPA PET imaging, and simple, off-the-shelf tests of working memory capacity, trait impulsivity, sEBR, and subjective reward sensitivity. Bayes factor analyses provided evidence in favour of the absence of correlations, except for the BAS questionnaire, for which the evidence was inconclusive. Furthermore, cross-validation analyses revealed little out-of-sample predictive power of each trait measure for dopamine synthesis capacity.

This work provides a direct assessment using human brain PET imaging of the relationships between these trait measures and striatal dopamine synthesis capacity on a much larger scale than previous work. Despite this, the strength of the correlations that we observed was far below the threshold of what we could reliably detect with an acceptable level of statistical power, should a true correlation exist. However, our sample size was more than adequate to reliably detect the effects of magnitudes that were previously observed, as the previous smaller-scale studies reported coefficients of 0.6 or higher for the correlations between dopamine synthesis capacity and working memory capacity (*Cools et al., 2008*; *Landau et al., 2009*) and impulsive behaviour (*van Holst et al., 2018*), as well as between direct measures of caudate nucleus dopamine levels and sEBR (*Taylor et al., 1999*). Instead, our results corroborate the findings of a lack of significant correlations between striatal dopamine synthesis capacity and working memory capacity (*Braskie et al., 2011*; *Braskie et al., 2008*; *Klostermann et al., 2012*), self-reported trait impulsivity (*van Holst et al., 2018*), and sEBR (*Sescousse et al., 2018*). Moreover, our Bayesian analyses demonstrated not just a lack of evidence for the presence of correlations, but also the presence of evidence for the absence of correlations. Nevertheless, genuine correlations may exist, undetected by this study, for example, because our relatively young and highly educated participant sample may not be representative enough of the general population. While uncovering such potential relationships would be relevant for our understanding of the links between striatal dopamine and trait characteristics, it seems unlikely those correlations would be strong enough to validate use of the trait measures as approximations of striatal dopamine function if they were not detected with the current participant sample.

One potential source of variability between the different PET studies of dopamine synthesis capacity is the use of different radiotracers. For example, this study used [$^{18}$F]-FDOPA to index dopamine synthesis capacity, whereas the previous studies reporting a positive correlation between working memory and striatal dopamine synthesis capacity used [$^{18}$F]-FMT (*Cools et al., 2008*; *Landau et al., 2009*). The tracer [$^{18}$F]-FDOPA is more commonly used and has been shown to have good test–retest reliability (*Egerton et al., 2010*; *Vingerhoets et al., 1994*), but the signal-to-noise ratio is greater for [$^{18}$F]-FMT because [$^{18}$F]-FDOPA is subject to additional in vivo COMT metabolism (*DeJesus et al., 1997*). That also means that the [$^{18}$F]-FDOPA signal reflects dopamine turnover to some degree rather than purely synthesis capacity (*Dejesus et al., 2001*), although the impact of dopamine turnover on the signal is only significant when longer scan times are used than we have presently used (*Sossi et al., 2001*). Future work is required to reconcile the growing body of literature demonstrating differential, sometimes even contrasting the effects of dopamine synthesis capacity measured with [$^{18}$F]-FMT and [$^{18}$F]-FDOPA (*Berry et al., 2016*; *Berry et al., 2018*; *Cools et al., 2009*; *Ito et al., 2011*; *Kumakura et al., 2010*; *van den Bosch et al., 2022*). While we cannot exclude the possibility, we consider it unlikely that the use of different radiotracers can fully explain the present lack of correlations as opposed to previous findings because there is similar discrepancy between [$^{18}$F]-FMT studies investigating working memory capacity, with some reporting positive correlations (*Cools et al., 2008*; *Landau et al., 2009*) and others reporting no effects or negative effects (*Braskie et al., 2011*; *Braskie et al., 2008*; *Klostermann et al., 2012*).

Critically, the absence of correlations was not due to measurement error in the estimates of striatal dopamine synthesis capacity or a lack of behavioural relevance of these estimates. The striatal [$^{18}$F]-FDOPA uptake values are in the typical range (*Figure 1—figure supplement 6*), and dopamine synthesis capacity in this dataset has been shown to account for (dopamine drug effects on) effort-based decision-making, response vigour, reversal learning, and smartphone social activity in the expected direction (*Hofmans et al., 2020*; *Hofmans et al., 2022*; *van den Bosch et al., 2022*; *Westbrook et al., 2020*; *Westbrook et al., 2021*). More specifically, [$^{18}$F]-FDOPA uptake in the striatum was associated positively with the value of cognitive effort (*Hofmans et al., 2020*; *Westbrook et al., 2020*; both with N = 46), with the impact of average reward value on response vigour (*Hofmans et al., 2022*; N = 44) and with dopaminergic drug-related changes in prediction error-related BOLD signal in the striatum (*van den Bosch et al., 2022*; N = 85). The implication of this body of work is that more sophisticated and quantitative indices of value-based learning, motivation, and even daily logs of participants' social activity on their smartphone (*Westbrook et al., 2021*; N = 22) might be better proxy measures of striatal dopamine synthesis capacity than the simple trait measures reported here. This is perhaps not surprising because these more sophisticated measurements provide much more detailed characterizations of (latent) biases measured over many trials, and in the case of the smartphone logs, over many days, rather than single (self-report) measurements of more stable trait measures. Possibly, a more sophisticated trial-wise approach of measuring sEBR during task performance might relate more strongly to (fluctuations in) striatal dopamine activity, given that it correlates with cognitive performance (*Ortega et al., 2022*). Nevertheless, the predictive value of even these more sophisticated measurements should be established using replication and/or cross-validation of the models in previously unseen data.

It should be noted that the present results speak specifically of the relationship of the trait measures with striatal dopamine synthesis capacity. They do not speak of the presence or absence of correlations with other aspects of the dopamine system, such as dopamine $D_{2/3}$-receptor availability or dopamine release. Indeed, working memory, trait impulsivity, and sEBR have all been associated with $D_{2/3}$-receptor availability and/or dopamine release (*Bäckman et al., 2011*; *Buckholtz et al., 2010*; *Clatworthy et al., 2009*; *Garrett et al., 2022*; *Groman et al., 2014*; *Jongkees and Colzato, 2016*; *Kim et al., 2014*), although the results for impulsivity and sEBR are mixed (*Dang et al., 2017*; *Lee et al., 2009*). Recent large-scale studies have demonstrated crucial involvement of both striatal and extrastriatal dopamine $D_{2/3}$-receptor availability in working memory performance using PET-fMRI measures (*Garrett et al., 2022*; *Salami et al., 2019*), but only prefrontal and not striatal receptor availability was significantly different between low and normal working memory groups (*Salami et al., 2018*). In addition to PET measures of striatal dopamine synthesis capacity, receptor availability, or dopamine release, future work might focus on other molecular imaging techniques and targets, such as single-photon emission computed tomography (SPECT) with ligands that bind to the dopamine transporter or prefrontal dopamine receptors. One such large-scale SPECT study in healthy volunteers (N = 188) did not find evidence for significant relationships between striatal dopamine transporter availability and working memory performance or trait impulsivity, in line with the current results for striatal dopamine synthesis capacity (*Burke et al., 2011*).

On a more fundamental level, future work is needed to clarify how the various PET imaging measures that index different aspects of dopamine system activity relate to each another. Investigating multiple aspects of the dopamine system is challenging as it requires different radiotracers and separate PET scans. Therefore, they are rarely studied within the same individuals. Nevertheless, a few small-scale studies have been conducted. In one such study, striatal dopamine synthesis capacity, as indexed by L-[$\beta$-$^{11}$C]-DOPA PET imaging, was found to correlate negatively with dopamine $D_{2/3}$-receptor availability, as indexed with [$^{11}$C]raclopride (*Ito et al., 2011*; N = 14), but no relationship was observed in others (*Heinz et al., 2005*; *Kienast et al., 2008*; *Yamamoto et al., 2021*; N = 24, 12, and 29, respectively). Conversely, another recent study (N = 40) that used [$^{18}$F]-FMT found striatal dopamine synthesis capacity to be positively correlated with striatal dopamine $D_{2/3}$-receptor availability but not with striatal dopamine release (*Berry et al., 2018*). If dopamine synthesis capacity is detached from actual amounts of dopamine released, then it may be less surprising that it does not show direct correlations with constructs that are impacted by dopamine within a healthy population. While dopamine synthesis capacity has been associated with a host of cognitive functions and tasks,

understanding the mechanisms behind these effects will ultimately require a deeper understanding of how the various aspects of the dopamine system interact.

The trait measures of working memory capacity, trait impulsivity, and sEBR have frequently been found to account for individual differences in the cognitive effects of dopaminergic drugs, for example, in the domain of reinforcement learning and Pavlovian biasing of instrumental action (*Cavanagh et al., 2014*; *Cools et al., 2003*; *Cools et al., 2007*; *Frank and O'Reilly, 2006*; *Froböse et al., 2018*; *Rostami Kandroodi et al., 2021*; *Kimberg et al., 1997*; *Mehta et al., 2000*; *van der Schaaf et al., 2014*). Such trait-dependent effects of dopamine drugs are reminiscent of the classic rate dependency hypothesis of psychostimulant effects (*Dews, 1977*), which states that manipulation effects depend on the baseline activity state of the system. We emphasize that the present results do not invalidate these previous findings but raise the hypothesis that the interactive effects between dopamine drug effects and trait characteristics reflect other factors than individual variation in baseline dopamine synthesis capacity. For example, dopaminergic drugs might have different effects on reinforcement learning in people with low and high working memory capacity because of differences in the degree to which the drugs boost reinforcement learning or working memory strategies for task performance (*Collins, 2018*; *Collins et al., 2014*).

In conclusion, the generally assumed link between baseline dopamine levels and the currently investigated trait measures has motivated popular use of these measures as a proxy for striatal dopamine activity instead of costly direct assessments of dopamine (synthesis) function with PET measurements. While such links are supported by evidence from patient or genetic studies and experimental animals, the interpretation of these behavioural indices as proxies for striatal dopamine levels requires robust direct evidence in humans for the existence of at least a strong correlation. The absence of such correlations observed here provides a strong cautionary message for future studies that may be tempted to use these relatively simple behavioural indices of working memory capacity, trait impulsivity, sEBR, or subjective reward sensitivity as a proxy for striatal dopamine synthesis capacity.

## Methods
### Participants
This study is part of a larger project (*Määttä et al., 2021*) for which 100 healthy volunteers were recruited, 50 women and 50 men (age at inclusion: range 18–43, mean [SD] = 23.0 [5.0] y). The sample size was determined based on the overarching project's aim to detect individual differences in dopaminergic drug effects as a function of dopamine synthesis capacity rather than the investigation of the relationships between dopamine synthesis capacity and putative proxy measures that we report here. All participants provided written informed consent and were paid 309 euros after completion of the overarching study. The study was approved by the local ethics committee ('Commissie Mensgebonden Onderzoek', CMO region Arnhem-Nijmegen, The Netherlands: protocol NL57538.091.16). People were recruited via an advertisement on the Radboud University electronic database for research participants (77%), a similar national database (https://www.proefbunny.nl/; 5%; a participant recruiting website that is not available anymore), and advertisement flyers around Nijmegen or word of mouth (together 18%). Prerequisites for participation were age between 18 and 45 y, Dutch as native language, and right-handedness. Before admission to the study, participants were extensively screened for adverse medical and psychiatric conditions. Exclusion criteria included any current or previous psychiatric or neurological disorders, having a first-degree family member with a current or previous psychiatric disorder, clinically significant hepatic, cardiac, renal, metabolic, or pulmonary disease, epilepsy, hyper- or hypotension, habitual smoking or drug use, pregnancy, and MRI contraindications, such as unremovable metal parts in the body or claustrophobia.

Six participants dropped out before completion of the study because of discomfort in the MRI or PET scanner (N = 4), personal reasons (N = 1), or technical failure of the PET scanner (N = 1). Therefore, PET data were available for a total of N = 94 participants (*Supplementary file 1* lists participant characteristics for this sample). For the sEBR analyses, one participant was excluded due to poor data quality and one was excluded as an outlier (more than 5 SDs from the group mean), resulting in a final sample of N = 92. For the impulsivity analyses, 67 participants completed the BIS-11 questionnaire (see subsection 'Trait impulsivity' for details), one of whom was excluded due to missing PET data,

resulting in a final sample of N = 66. For the Digit Span, Listening Span, and BAS questionnaire, the full sample of N = 94 was included.

## General procedure

Data were collected as part of a large PET, pharmaco-fMRI study on the effects of methylphenidate and sulpiride on brain and cognition, employing a within-subject, placebo-controlled, double-blind cross-over design (Netherlands Trial Register 5959; https://trialsearch.who.int/Trial2.aspx?TrialID= NTR6140). For a detailed description of the testing sessions and tasks and measures collected, see *Määttä et al., 2021*.

The study consisted of five testing days separated by at least 1 wk. The first was an intake session in which participants were screened for inclusion criteria, an anatomical MRI scan was obtained, and the measures of sEBR, and Listening Span and Digit Span of working memory capacity were collected. The second, third, and fourth testing days were 6-hr-long pharmaco-fMRI sessions in which participants performed a battery of dopamine-related tasks not reported here. On the fifth day, participants performed the Digit Span test once more, completed two behavioural tasks, and finally underwent an [$^{18}$F]-FDOPA PET scan of the brain to measure their dopamine synthesis capacity.

Participants filled in the questionnaires of personality and trait characteristics online, after the third pharmaco-fMRI session and before the final testing day with PET scan (mean time difference = 32.45 d; standard error of mean = 3.14 d). The mean time difference between the intake session and PET session was 90.16 d, standard error of mean = 3.13 d.

## Data acquisition and preprocessing

### MRI

On the intake session, a whole-brain structural image was acquired to use for within-subject registration with the PET images, using a T1-weighted magnetization prepared, rapid-acquisition gradient echo sequence (192 sagittal slices; repetition time, 2300 ms; echo time, 3.03 ms; field of view: 256 × 256 mm; flip angle, 8°; 256 × 256 matrix; 1.0 mm in-plane resolution; 1.0 mm slice thickness). The MRI experiment was performed on a 3T Siemens Magnetom Skyra MRI scanner at the Donders Institute using a 32-channel head coil.

### PET

The brain PET data were acquired on a state-of-the-art PET/CT scanner (Siemens Biograph mCT; Siemens Medical Systems, Erlangen, Germany) at the Department of Medical Imaging of the Radboud University Medical Center. We used the well-validated radiotracer [$^{18}$F]-FDOPA, which was synthesized at Radboud Translational Medicine BV (RTM BV) in Nijmegen. The tracer is a substrate for aromatic amino acid decarboxylase, the enzyme that converts DOPA into dopamine. The rate of conversion of [$^{18}$F]-FDOPA into dopamine provides an estimate of dopamine synthesis capacity. It is a stable measure with good test–retest reliability (intraclass correlation coefficients range from about 0.7–0.94 in the striatum) even after a 2-yr time interval between acquisitions of scans (*Egerton et al., 2010*; *Vingerhoets et al., 1994*). 50 min before the PET scan started, participants received 150 mg of carbidopa and 400 mg of entacapone to minimize peripheral metabolism of [$^{18}$F]-FDOPA by peripheral decarboxylase and catechol-O-methyltransferase (COMT), respectively, thereby increasing signal-to-noise ratio in the brain (*Boyes et al., 1986*; *Hoffman et al., 1992*; *Ishikawa et al., 1996*; *Léger et al., 1998*).

The procedure started with a low-dose CT scan to use for attenuation correction of the PET images. Then, the [$^{18}$F]-FDOPA tracer was administered (approximately 185 MBq) via a bolus injection in the antecubital vein and the PET scan was started. Dynamic PET data (4 × 4 × 3 mm voxel size; 5 mm slice thickness; 200 × 200 × 75 matrix) were acquired over 89 min and divided into 24 frames (4 × 1, 3 × 2, 3 × 3, 14 × 5 min). Data were reconstructed with weighted attenuation correction and time-of-flight recovery, scatter corrected, and smoothed with a 3 mm full-width-at-half-maximum (FWHM) kernel.

The PET data were preprocessed and analysed using SPM12. All frames were realigned to the mean image to correct for head motion between scans. The realigned frames were then co-registered to the structural MRI scan using the mean PET image of the first 11 frames (corresponding to the first 24 min), which has a better range in image contrast than a mean image over the whole scan time in regions other than the striatum. Presynaptic dopamine synthesis capacity was quantified as the tracer

**Table 3.** Pearson correlation coefficients for dopamine synthesis capacity in the three striatal regions of interest.

| | Caudate nucleus | Putamen | Nucleus accumbens |
|---|---|---|---|
| Caudate nucleus | 1 | 0.751 | 0.649 |
| Putamen | 0.751 | 1 | 0.787 |
| Nucleus accumbens | 0.649 | 0.787 | 1 |

influx rate $k_i^{cer}$ (min$^{-1}$) per voxel with graphical analysis for irreversible tracer binding using Gjedde–Patlak modelling (*Patlak and Blasberg, 1985*; *Patlak et al., 1983*). The analysis was performed on the images corresponding to 24–89 min, which is the period after the irreversible compartments had reached equilibrium and the input function to the striatum had become linear. The $k_i^{cer}$ values represent the rate of tracer accumulation relative to the reference region of cerebellar grey matter, where the density of dopamine receptors and metabolites is extremely low compared to the striatum (*Farde et al., 1986*; *Hall et al., 1999*). The cerebellar grey matter mask was obtained using FreeSurfer segmentation of each individual's anatomical MRI scan, as implemented in *fMRIPREP*. The resulting $k_i^{cer}$ maps were spatially normalized to MNI space, smoothed with an 8 mm FWHM kernel and brain extracted.

After preprocessing, we extracted the mean $k_i^{cer}$ values from the caudate nucleus, putamen, and nucleus accumbens in native subject space. The masks for these ROIs were taken from an independent parcellation of the striatum based on intra-striatal functional connectivity (*Figure 1—figure supplement 7*; *Piray et al., 2017*). The extracted $k_i^{cer}$ values represent the dopamine synthesis capacity in the ROIs and were used to correlate the putative proxy measures with. There were strong correlations between dopamine synthesis capacity values in the three ROIs (*Table 3*).

## Working memory

Working memory capacity was indexed with the Digit Span test (*Groth-Marnat, 1997*) and Listening Span test (*Daneman and Carpenter, 1980*). Both tasks were computerized.

In the Digit Span test, participants started with the forward section, in which they listened to a series of numbers and were required to repeat the correct numbers in order. The series increased in length (starting at three digits) until the participant failed two attempts on a series length, or until the maximum length of nine digits was completed. Next, in the backwards section, participants again listened to a series of numbers but were required to repeat the correct numbers in reverse order. This series started with a length of two digits and increased until the participant failed two attempts on a series length, or until a maximum length of eight digits. The Digit Span score on each section was the number of correctly recited series, and the total score was the sum of the scores on the forward and backward sections. The Digit Span was performed twice, once on the intake session and once on the final testing day, before the PET scan (test–retest correlation: $\rho$ = 0.8). The scores on each day were averaged to obtain the final Digit Span scores for the forward and backward sections, as well as the total scores.

The Listening Span test consists of sets of pre-recorded sentences, increasing in size from two to seven sentences. Participants were presented with the sentences and were simultaneously required to answer written verification questions regarding the content of each sentence. At the end of each set, subjects recalled the final word of each sentence in the order of presentation. The Listening Span was defined as the set size for which the participant correctly recalled the final words on at least two out of three trials. Listening span increased with half a point when only one trial of the next level was correct. In addition to the Listening Span, the total number of final words that were correctly recalled was used as an additional measure.

## Trait impulsivity

Trait impulsivity was assessed with the Barratt Impulsiveness Scale (BIS-11; *Patton et al., 1995*). The BIS-11 is a self-report questionnaire, consisting of 30 statements about common (non)impulsive behaviours and preferences, and participants indicate their level of endorsement on a 4-point

Likert scale. The BIS-11 total impulsivity scores reflect the tendency towards impulsivity. Participants completed the questionnaire at home between test days.

Due to an experimenter error, a prototype of the questionnaire, known as BIS-11A, was originally administered. That version has never been validated and contains both differently phrased as well as completely different questions compared with the validated BIS-11 questionnaire (https://www.impul-sivity.org/measurement/bis11/). When the error was discovered after the completion of the study, the intended version of the questionnaire (i.e. BIS-11) was sent to the participants, of whom 67 completed it.

### sEBR

The sEBR data were collected using electro-oculography (EOG), recorded on a BrainVision Recorder system (Brain Products GmbH, Munich, Germany) with 200 Hz sampling rate. We followed automatic and manual procedures for data acquisition, preprocessing, and data analysis of the EOG data (*Sescousse et al., 2018*). The EOG data were acquired during the day (before 5 pm) over a period of (up to) 10 min using two vertical and two horizontal Ag-AgCl electrodes placed around the eyes. During the study, the length of recording was increased from 6 to 10 min following recommendation by external colleagues; 28 participants were recorded for 6 min, and the 72 remaining participants were recorded for 10 min. The vertical EOG (vEOG) signal, used for assessing eye blinks, was obtained from a bipolar montage using the electrodes placed above and below the right eye. The horizontal EOG (hEOG) signal, used to exclude artefacts produced by saccades and muscle activity, was obtained from bipolar montage using the electrodes placed lateral to the eternal canthi, and the ground electrode was placed on the right mastoid.

Participants were comfortably seated facing a white wall from about 1.5 m distance and were asked to look ahead while they believed the experimenter was checking signal quality in preparation of a system calibration. The measurement took place during what the participants believed was a signal quality check. This cover story was meant to prevent the participants from being aware that they were being recorded. The experimenter was outside the room during the measurement. No instructions about blinking were given, and care was taken to not direct participants' attention to their eyes. However, due to a communication error, nine participants received inconsistent instructions and were incorrectly instructed to stare at the wall. Excluding these participants slightly increased the coefficients of the correlations with dopamine synthesis capacity, but not enough to become statistically significant (see 'Results).

The EOG data were rectified and band-pass filtered between 0.5 and 20 Hz using the Fieldtrip toolbox in MATLAB (*Oostenveld et al., 2011*; http://fieldtriptoolbox.org/; MathWorks Inc; https://nl.mathworks.com/products/matlab.html). Eye blinks were detected using an automated procedure based on a voltage change passing a threshold that was set individually per participant (range 100 µV) in a time interval of 400 ms (*Slagter et al., 2010*). The vEOG signal was visually inspected by two researchers independently to assess detection accuracy. Blinks were manually added or removed according to the threshold definition of a blink when appropriate, and potential artefacts from saccades or muscle activity were removed when they were detected in the hEOG signal. Because the inter-rater reliability of the independent scoring of the sEBR values was high ($\alpha = 0.98$), the analyses were performed on the average blink rate determined by the two researchers.

### Subjective reward sensitivity

We measured trait characteristics of reward sensitivity with the Behavioural Activation Scale (BAS; *Carver and White, 1994*; *Franken et al., 2005*). This scale consists of 17 items that are scored on a 4-point Likert scale and can be split into three subscales: Reward-Responsiveness, Drive, and Fun-Seeking. These were designed to assess the personality constructs of reinforcement sensitivity theory (*Carver and White, 1994*).

## Statistical analysis

The primary correlation analyses were performed with dopamine synthesis capacity for each of the three striatal ROIs and the total Digit Span score and Listening Span score, total BIS-11 score, sEBR values, and total BAS score. We subsequently explored the correlations between dopamine synthesis capacity and sub-sections of the proxy measures: forward and backward Digit Span, total words

recalled in the Listening Span task, impulsivity scores on the attentional, motor and nonplanning subscales of the BIS-11, and scores on the Reward Responsiveness, Fun-Seeking and Drive subscales of the BAS questionnaire. Statistics for the correlations with the subscales are provided in *Figure 1—figure supplements 1–3* but not used for inference on statistical significance, given the exploratory nature of these analyses.

All statistical analyses were performed in R (version 4.0.1; *R Development Core Team, 2020*). We used one-sided Pearson correlation analyses to test the statistical significance of positive relationships between the proxy measures and dopamine synthesis capacity in the three striatal ROIs. For each proxy, we corrected the p-values of the correlation coefficients for the comparisons in three ROIs using Holm–Bonferroni correction. Since many of the corrected p-values were rounded to the upper limit of 1, we also report the uncorrected values to give a sense of precision.

For a more precise localization of any correlation effects, we also performed voxel-wise one-sample *t*-tests of the PET $k_i^{cer}$ data with each proxy measure as a covariate in separate models. We report striatal activation results for clusters surviving peak-level family-wise error (FWE) correction at $p<0.05$ after small-volume correction for the combination of the three striatal ROI masks. These analyses were run using Statistical Parametric Mapping 12 (SPM12; https://www.fil.ion.ucl.ac.uk/spm/software/spm12/) running in MATLAB.

In addition to the confirmatory Pearson correlation tests, we used Bayes factor correlation analyses to quantify the evidence in favour of the null hypothesis (H0) that there is not a positive correlation versus the alternative hypothesis (H1) that there is a positive correlation. The Bayes factor ($BF_{01}$) reports the ratio of the evidence for H0 over H1, given the data. Thus, a $BF_{01}$ greater than 1 indicates stronger evidence for H0 than H1, and vice versa for a $BF_{01}$ smaller than 1. We ran the Bayesian correlation analyses with uninformative (flat) prior beta distributions over the (0,1) interval. We also checked the robustness of the resulting Bayes factors by recalculating the Bayes factors for each correlation using a range of strong to weak beta priors (ranging from a width of 0–2). For completeness, we also quantified the evidence for the non-directional hypotheses of no correlation (H0) versus a non-zero correlation (H1) using Bayesian correlation analyses over the (–1,1) interval with uninformative priors. The Bayes factor analyses were performed using the function `correlationBF` of the `BayesFactor` package in R (*Ly et al., 2016*; version 0.9.12–4.2).

## Predictive modelling

We used *k*-fold cross-validation and permutation testing to estimate to what extent the trait variables could be used to predict dopamine synthesis capacity for previously unseen participants. These resampling methods were applied to simple linear regression models, in which dopamine synthesis capacity in one of the three striatal ROIs served as the outcome variable and one of the trait measures served as the predictor variable.

In *k*-fold cross-validation, the original data is randomly split into *k* subsamples (i.e. folds) of roughly equal size. The model is initially fit to a 'training' dataset comprising *k*-1 folds. The trained model is then applied to the single remaining fold that was held back as the 'validation' data to evaluate the predictive accuracy. This process is repeated *k* times, so that each fold is used exactly once as the validation data. The entire *k*-fold cross-validation procedure can be repeated several times to avoid spurious results due to the random initial split into *k* folds. Here, we performed tenfold cross-validation with 100 repeats.

Permutation sampling is used to construct a null distribution of the model's predictive accuracy, which can be compared to the estimated predictive accuracy for the original data to establish whether the model offers predictive performance above chance. For a single permutation, the outcome variable is randomly shuffled whereas the predictor variable is unaltered, thereby eliminating the participant-wise link between variables. This permuted dataset is then subjected to tenfold cross-validation to estimate the model's predictive accuracy. By repeating this process many times (here, 5000 times), we obtain a distribution of the model's predictive accuracy under the null hypothesis. The permutation p-value is defined as the proportion of permutations for which the model's predictive accuracy is as good or greater than the predictive accuracy for the original data.

We defined predictive accuracy as the model's coefficient of determination, $R^2$. If $y_i$ is the predicted value of the *i*-th sample and $y_i$ is the corresponding true value for a total of *n* samples, the estimated $R^2$ is defined as

$$R^2(y, \hat{y}) = 1 - \frac{\sum_{i=1}^{n}\left(y_i - \hat{y}_i\right)^2}{\sum_{i=1}^{n}\left(y_i - \bar{y}\right)^2}$$

where $\bar{y} = \frac{1}{n}\sum_{i=1}^{n}y_i$ . An $R^2$ of 1 indicates perfect predictive accuracy. A constant (intercept-only) model that merely predicts the expected value of $y$, ignoring individual differences as a function of the predictor variable, would correspond to an $R^2$ of 0. The model can be arbitrarily worse than an intercept-only model, and therefore $R^2$ can be negative.

We additionally examined the root mean square error, RMSE, which represents the quadratic mean of the difference between the true values and predicted values:

$$\text{RMSE} = \sqrt{\frac{1}{n}\sum_{i=1}^{n}(y_i - \hat{y}_i)^2}$$

In contrast to $R^2$ , this metric is represented on the same scale as the outcome variable. An RMSE of 0 indicates perfect predictive accuracy, and higher scores correspond to worse accuracy.

## Acknowledgements

We thank Margot van Cauwenberge, Peter Mulder, and Monique Timmer for medical assistance during data collection, and we also thank the people that participated in this study. The work was funded by a Vici grant to RC from the Netherlands Organization for Scientific Research (NWO; grant no. 453-14-015). This project has received a Voucher from the European Union's Horizon 2020 Framework Programme for Research and Innovation under the Specific Grant Agreement No. 945539 (Human Brain Project SGA3).

## Additional information

### Funding

| Funder | Grant reference number | Author |
| --- | --- | --- |
| Nederlandse Organisatie voor Wetenschappelijk Onderzoek | 453-14-015 | Roshan Cools |
| Horizon 2020 Framework Programme | 945539 | Roshan Cools |

The funders had no role in study design, data collection and interpretation, or the decision to submit the work for publication.

### Author contributions

Ruben van den Bosch, Data curation, Software, Formal analysis, Validation, Investigation, Visualization, Writing - original draft, Project administration, Writing – review and editing; Frank H Hezemans, Software, Formal analysis, Validation, Visualization, Writing - original draft, Writing – review and editing; Jessica I Määttä, Data curation, Investigation, Project administration, Writing – review and editing; Lieke Hofmans, Danae Papadopetraki, Robbert-Jan Verkes, Investigation, Writing – review and editing; Andre F Marquand, Jan Booij, Writing – review and editing; Roshan Cools, Conceptualization, Supervision, Funding acquisition, Project administration, Writing – review and editing

### Author ORCIDs
Ruben van den Bosch ⓘ http://orcid.org/0000-0002-3994-8291

## Ethics

Human subjects: All participants provided written informed consent. The study was approved by the local ethics committee ("Commissie Mensgebonden Onderzoek", CMO region Arnhem-Nijmegen, The Netherlands: protocol NL57538.091.16).

## Decision letter and Author response

Decision letter https://doi.org/10.7554/eLife.83161.sa1
Author response https://doi.org/10.7554/eLife.83161.sa2

---

## Additional files

### Supplementary files
- MDAR checklist
- Supplementary file 1. Participant characteristics (N = 94).

### Data availability

The minimally processed data used in this study and the overarching project it is part of are available from the Donders Institute Data Repository (https://doi.org/10.34973/wn51-ej53; custom data use agreement RU-DI-HD-1.0). The final data derivatives relevant to the current work, as well as all code for data analysis and figures creation, are available from a separate collection on the Donders Institute Data Repository (https://doi.org/10.34973/0sce-z290).

The following datasets were generated:

| Author(s) | Year | Dataset title | Dataset URL | Database and Identifier |
|---|---|---|---|---|
| Määttä JIM, Cools R, van den Bosch R, Hofmans L, Papadopetraki D, Westbrook A, Lambregts BIHM | 2020 | Effects of sulpiride and methylphenidate on brain and cognition: a PET pharmaco-fMRI study | https://doi.org/10.34973/wn51-ej53 | Donders Institute Data Repository, 10.34973/wn51-ej53 |
| van den Bosch R, Hezemans F, Määttä JI, Hofmans L, Papadopetraki D, Verkes RJ, Marquand A, Booij J, Cools R | 2023 | Evidence for absence of links between striatal dopamine synthesis capacity and working memory capacity, spontaneous eye-blink rate, and trait impulsivity | https://doi.org/10.34973/0sce-z290 | Donders Institute Data Repository, 10.34973/0sce-z290 |

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
