## [Editor Report]

This study presents fundamental insights into the relationship between [18F]-FDOPA PET measurements of striatal dopamine synthesis capacity and a series of measures, including behavioral readouts, proposed to index dopamine function. The frequentist and Bayesian analyses together provide compelling evidence for an absence of any relationship between striatal dopamine synthesis capacity and external measures, questioning the interpretation of studies using such measures to index dopamine function. These findings will not only be of great interest to cognitive neuroscientists but also inform future studies of neuropsychiatric diseases.

---

## [Decision Letter]

**Decision letter after peer review:**

Thank you for submitting your article "Evidence for absence of links between striatal dopamine synthesis capacity and working memory capacity, spontaneous eye-blink rate, and trait impulsivity" for consideration by *eLife*. Your article has been reviewed by 2 peer reviewers, and the evaluation has been overseen by a Reviewing Editor and Christian Büchel as the Senior Editor. The following individuals involved in the review of your submission have agreed to reveal their identity: Mitul Mehta (Reviewer #1); David Zald (Reviewer #2).

Essential revisions:

1) The study is seated in the empirical use of 'baseline' assessments and the theoretical and mechanistic plausibility of this position is only briefly highlighted in the introduction. The use of the terms 'linked' and 'link' are handy but can hide a more complex relationship that may be non-linear. It would be great if you could expand a little here on the precise relationships which have been suggested and described previously.

2) The study is focused on striatal dopamine assessment whereas in the introduction subcortical and cortical dopamine is discussed. A much clearer separation might be helpful here, arguably dismissing the cortical element briefly and leaving this for the discussion, with a more critically evaluative assessment of the subcortical processes predicted to be mediated by dopamine transmission. In addition, the 'functional' subdivisions and their importance should be introduced.

3) Please provide more information on the group included. Multiple sources of recruitment are described. Please indicate the number included from each source. Please include a table of participant characteristics. Any measure of the social status of participants (socio-economic status, income band, etc), and ethnicity, would also be appreciated.

4) Please indicate the n for each of the DA synthesis studies mentioned in lines 107-113 (p. 6). Similarly in the discussion (lines 331-334), it is reasonable for a reader to wonder what the sample size was for studies of the value of cognitive effort (Westbrook et al., 2020), with the impact of average reward value on response vigor (Hofmans et al., 2022) and with dopaminergic drug-related changes in prediction error-related BOLD signal in the striatum. If these are of similarly small samples as the working memory studies it would suggest as much suspicion should be applied to these other findings.

5) Please discuss additional reasons why the presented index of dopamine synthesis might not prove informative. A discussion of how dopamine synthesis capacity relates to other dopaminergic parameters such as dopamine release warrants discussion given that the core hypothesis driving the study is that individual differences in synthesis capacity within the normal range are functionally impactful. If, for instance, dopamine synthesis capacity is detached from actual amounts of dopamine released, then it may be less surprising that it is not showing relations to constructs that are impacted by dopamine within a healthy population.

6) It is very much appreciated that the authors have provided information on the predictive power of the observed effect sizes. Given a negative finding, it may also be useful to articulate what statistical power the study had in the first place in relation to possible effect sizes (Cohen's D). Similarly, it would be useful to articulate the test-retest reliability of synthesis capacity metrics, since unreliability would naturally lower the ability to observe correlates of synthesis capacity, and thus alter statistical power.

*Reviewer #1 (Recommendations for the authors):*

I only have a small number of recommendations for the authors.

- The study is seated in the empirical use of 'baseline' assessments and the theoretical and mechanistic plausibility of this position is only briefly highlighted in the introduction. The use of the terms 'linked' and 'link' are handy but can hide a more complex relationship that may be non-linear. It is worth expanding a little here on the precise relationships which have been theorised and described previously. This speaks directly to what the authors are attempting to model.

- The study is focussed on striatal dopamine assessment whereas in the introduction subcortical and cortical dopamine is discussed. The authors should attempt a much clearer separation here, arguably dismissing the cortical element briefly and leaving this for the discussion, with a more critically evaluative assessment of the subcortical processes predicted to be mediated by dopamine transmission. In addition, the 'functional' subdivisions and their importance need to be introduced.

- The authors do not provide sufficient information on the group included. Multiple sources of recruitment are described. Please indicate the number included from each source. Please include a table of participant characteristics. If you have any measure of the social status of participants (socio-economic status, income band, etc), and ethnicity, please also include this.

Congratulations on this work.

*Reviewer #2 (Recommendations for the authors):*

Because the methods are sound and the paper is well-written, my comments are relatively limited. Nevertheless, there are a few things that might further improve the paper.

1) Since a major argument for this study is that many of the existing papers on the topic have small samples relative to the current one, on page 6 it would be useful to indicate the n for each of the DA synthesis studies mentioned in lines 107-113. Similarly in the discussion (lines 331-334), it is reasonable for a reader to wonder what the sample size was for studies of the value of cognitive effort (Westbrook et al., 2020), with the impact of average reward value on response vigor (Hofmans et al., 2022) and with dopaminergic drug-related changes in prediction error-related BOLD signal in the striatum, since if these are of similarly small samples as the working memory studies it would suggest as much suspicion should be applied to these other findings.

2) The authors may wish to discuss if there are additional reasons why their index of dopamine synthesis might not prove informative. They note in the discussion that F-Dopa is subject to dopamine turnover. This is true, and of course, could potentially be measured with the proper methodology, so it is probably worth an explanation of why they do not specifically model turnover. The issue of low signal-to-noise ratios for their FDOPA measures that they raise in the discussion seems a less likely culprit and hardly seems worth describing. However, a consideration of how dopamine synthesis capacity relates to other dopaminergic parameters such as dopamine release warrants discussion given that the core hypothesis driving the study is that individual differences in synthesis capacity within the normal range are functionally impactful. If, for instance, dopamine synthesis capacity is detached from actual amounts of dopamine released, then it may be less surprising that it is not showing relations to constructs that are impacted by dopamine within a healthy population.

3) I appreciate that the authors have provided information on the predictive power of the observed effect sizes. Given a negative finding, it may also be useful to also articulate what statistical power the study had in the first place in relation to a small or moderate effect size (Cohen's D). Similarly, it would be useful to articulate the test-retest reliability of synthesis capacity metrics, since unreliability would naturally lower the ability to observe correlates of synthesis capacity, and thus alter statistical power.

---

## [Author Response]

Essential revisions:1) The study is seated in the empirical use of 'baseline' assessments and the theoretical and mechanistic plausibility of this position is only briefly highlighted in the introduction. The use of the terms 'linked' and 'link' are handy but can hide a more complex relationship that may be non-linear. It would be great if you could expand a little here on the precise relationships which have been suggested and described previously.

We have now expanded the precise observations of previous studies regarding the relationships between the trait variables and striatal dopamine. With this, we have replaced instances of our use of the term ‘link’ with more explicit statements like ‘positive correlation’.

More specifically, for working memory we explain that greater caudate nucleus dopamine synthesis capacity and release has been associated with greater working memory capacity and that this prior observation pertained in particular to the updating of working memory. For impulsivity, we now open the paragraph with the key findings of positive correlations between impulsivity and human PET indices of dopamine function, rather than starting with another instance of the word ‘link’. In addition, we have moved the two animal study references more appropriately to the counter-evidence section and decided to include here some compelling evidence from studies on Parkinson’s disease. For sEBR, we have rephrased the paragraph, keeping the explicit mention of positive correlation findings, and have added the clause that the exact mechanism by which sEBR and striatal dopamine activity are connected remains unclear.

2) The study is focused on striatal dopamine assessment whereas in the introduction subcortical and cortical dopamine is discussed. A much clearer separation might be helpful here, arguably dismissing the cortical element briefly and leaving this for the discussion, with a more critically evaluative assessment of the subcortical processes predicted to be mediated by dopamine transmission. In addition, the 'functional' subdivisions and their importance should be introduced.

The introduction was in fact already focused on subcortical dopamine. We have now made this explicit by rewriting the opening paragraph and including a sentence stating our focus on striatal dopamine:

“The mesocorticolimbic dopamine system plays a key role in a range of cognitive functions, including cognitive control processes, such as working memory (Arnsten and Li, 2005), attention (Thiele and Bellgrove, 2018), and flexible behaviour (Floresco and Magyar, 2006). While such processes have classically been associated with prefrontal dopamine function, they also critically depend on the basal ganglia and dopamine activity in the striatum (Cools, 2019; Frank and O’Reilly, 2006; Ott and Nieder 2019). Accordingly, individual differences in striatal dopamine function have been associated with various behavioural and physiological trait characteristics, including working memory capacity, impulsivity, and spontaneous eye-blink rate (sEBR). We focused on striatal dopamine and investigated the relationships suggested by prior literature between these trait characteristics and individual variation in striatal dopamine function.”

In addition, we suspect that phrasings like “dopamine function” might be taken to mean dopamine processes in general, including cortical dopamine, which might have made the focus on specifically striatal dopamine less clear. Therefore, we have rephrased those instances to “striatal dopamine function”.

To better introduce the relevance of striatal subdivisions, as well as for a more precise discussion of previous findings, we have now also referred to the specific striatal locus of the observed relationships with dopamine within the striatum. At the end of the introduction we have added a section explaining the rationale and origin of our striatal ROIs:

“The subcortical dopamine system is not a single entity and the striatum is a functionally heterogeneous structure with a distinct connectionist anatomy with the cortex, involving a functional gradient in the connections between the cortex and ventral striatum (primarily nucleus accumbens), dorsolateral striatum (putamen), and dorsomedial striatum (caudate nucleus; Alexander et al., 1986; Haber et al., 2000; Joel and Weiner, 2000). To take this heterogeneity into account, we performed our analyses in three striatal regions of interest (ROIs), defined using a parcellation based on intra-striatal functional connectivity in an independent sample (Piray et al., 2017). The ROIs approximately matched the anatomical subdivision of the striatum into caudate nucleus, putamen, and nucleus accumbens (ventral striatum). We tested the hypotheses that the trait measures were positively correlated with, and predictive of, estimates of striatal dopamine synthesis capacity in these striatal ROIs.”

3) Please provide more information on the group included. Multiple sources of recruitment are described. Please indicate the number included from each source. Please include a table of participant characteristics. Any measure of the social status of participants (socio-economic status, income band, etc), and ethnicity, would also be appreciated.

18% of the participants approached us after seeing the advertisement on flyers or heard about the study via word of mouth, 5% found us on proefbunny.nl, the rest were recruited via the university’s electronic participant database. We have now added this information to the text in the Methods section. We have also added a table (Supplementary Table 1 in Supplementary File 1) with participants characteristics specifying the already provided information on sex and age distribution, and we also added education level and current occupation. However, we did not collect information on income or ethnicity.

4) Please indicate the n for each of the DA synthesis studies mentioned in lines 107-113 (p. 6). Similarly in the discussion (lines 331-334), it is reasonable for a reader to wonder what the sample size was for studies of the value of cognitive effort (Westbrook et al., 2020), with the impact of average reward value on response vigor (Hofmans et al., 2022) and with dopaminergic drug-related changes in prediction error-related BOLD signal in the striatum. If these are of similarly small samples as the working memory studies it would suggest as much suspicion should be applied to these other findings.

We have added the N for each of these studies in the introduction and discussion.

In the introduction: “(… all using subsets from one sample of N=37 participants, of which 20 overlapped with Landau et al., 2009).”

In the discussion: “More specifically, [^18^F]-FDOPA uptake in the striatum was associated positively with the value of cognitive effort (Hofmans et al., 2020; Westbrook et al., 2020; both with N=46), with the impact of average reward value on response vigour (Hofmans et al., 2022; N=44) and with dopaminergic drug-related changes in prediction error-related BOLD signal in the striatum (van den Bosch et al., 2022; N=85). The implication of this body of work is that more sophisticated and quantitative indices of value-based learning, motivation, and even daily logs of participants’ social activity on their smartphone (Westbrook et al., 2021; N=22) might be better proxy measures of striatal dopamine synthesis capacity than the simple trait measures reported here.”

5) Please discuss additional reasons why the presented index of dopamine synthesis might not prove informative. A discussion of how dopamine synthesis capacity relates to other dopaminergic parameters such as dopamine release warrants discussion given that the core hypothesis driving the study is that individual differences in synthesis capacity within the normal range are functionally impactful. If, for instance, dopamine synthesis capacity is detached from actual amounts of dopamine released, then it may be less surprising that it is not showing relations to constructs that are impacted by dopamine within a healthy population.

We do think that dopamine synthesis capacity is functionally impactful, as evidenced by many previously observed effects on behaviour and highlighted in another paragraph about findings in our current participant sample. However, to understand the mechanisms by which it impacts behaviour or function it is indeed important to better understand how dopamine synthesis capacity relates to other dopamine parameters. A detachment of dopamine synthesis capacity from release would indeed make it less surprising to not observe relationships with dopamine-related constructs. We agree that some additional discussion regarding the association between dopamine synthesis capacity and other dopamine parameters, including release, is key. We have now added a paragraph discussing this:

“On a more fundamental level, future work is needed to clarify how the various PET imaging measures that index different aspects of dopamine system activity relate to each another. Investigating multiple aspects of the dopamine system is challenging, as it requires different radiotracers and separate PET scans. Therefore they are rarely studied within the same individuals. Nevertheless, a few small-scale studies have been conducted. In one such study, striatal dopamine synthesis capacity, as indexed by L-[B-^11^C]-DOPA PET imaging, was found to correlate negatively with dopamine D_2/3_-receptor availability, as indexed with [^11^C]raclopride (Ito et al., 2011; N=14), but no relationship was observed in other studies (Heinz et al., 2005; Kienast et al., 2008; Yamamoto et al., 2021; N=24, 12, and 29, respectively). Conversely, another recent study (N=40) that used [^18^F]-FMT found striatal dopamine synthesis capacity to be positively correlated with striatal dopamine D_2/3_-receptor availability but not with striatal dopamine release (Berry et al., 2018). If dopamine synthesis capacity is detached from actual amounts of dopamine released, then it may be less surprising that it does not show direct correlations with constructs that are impacted by dopamine within a healthy population. While dopamine synthesis capacity has been associated with a host of cognitive functions and tasks, understanding the mechanisms behind these effects will ultimately require a deeper understanding of how the various aspects of the dopamine system interact.”

We have also rewritten the paragraph on the use of [^18^F]-FDOPA versus [^18^F]-FMT to index dopamine synthesis capacity and added an explanation of why we do not model dopamine turnover, even though [^18^F]-FDOPA is subject to dopamine turnover: “… the impact of dopamine turnover on the signal is only significant when longer scan times are used than we have presently used (Sossi et al., 2001).”

6) It is very much appreciated that the authors have provided information on the predictive power of the observed effect sizes. Given a negative finding, it may also be useful to articulate what statistical power the study had in the first place in relation to possible effect sizes (Cohen's D). Similarly, it would be useful to articulate the test-retest reliability of synthesis capacity metrics, since unreliability would naturally lower the ability to observe correlates of synthesis capacity, and thus alter statistical power.

We thank the reviewers for this suggestion. We have added these details to the end of the first Results section in the following paragraph:

“To further inspect the impact of sample size and power, we calculated the effect size that we would be able to reliably detect with the one-sided Pearson correlation tests at acceptable levels of statistical power in our sample of N=94 (with statistical significance level of α = 0.05), using G*Power (Faul et al., 2009). With power of 0.9 we would be able to reliably detect correlations with a coefficient of ρ = 0.29 (which corresponds to a sample Cohen’s d of 0.61). For a power of 0.8 the coefficients would have to be ρ = 0.25 (sample Cohen’s d = 0.52). The hypothesized correlations between striatal dopamine synthesis capacity and working memory capacity, impulsivity, and sEBR were all much weaker than that in the current sample: all but one correlation with ρ < 0.1. For an effect size of ρ = 0.1 and power of 0.8 a sample size of N=614 would have been needed (N=850 for power of 0.9).”

We also slightly rewrote the second paragraph of the Discussion to add a short discussion of these numbers, also including an acknowledgement of the potential existence of true correlations undetected by our sample:

“The current work provides a direct assessment using human brain PET imaging of the relationships between these trait measures and striatal dopamine synthesis capacity on a much larger scale than previous work. Despite this, the strength of the correlations that we observed was far below the threshold of what we could reliably detect with an acceptable level of statistical power, should a true correlation exist. However, our sample size was more than adequate to reliably detect effects of magnitudes that were previously observed, as the previous smaller-scale studies reported coefficients of 0.6 or higher for the correlations between dopamine synthesis capacity and working memory capacity (Cools et al., 2008; Landau et al., 2009), impulsive behaviour (van Holst et al., 2018), and sEBR (Taylor et al., 1999). Instead, our results corroborate findings of a lack of significant correlations between striatal dopamine synthesis capacity and working memory capacity (Braskie et al., 2008; Braskie et al., 2011; Klostermann et al., 2012), self-reported trait impulsivity (van Holst et al., 2018), and sEBR (Sescousse et al., 2018). Moreover, our Bayesian analyses demonstrated not just a lack of evidence for the presence of correlations, but also the presence of evidence for the absence of correlations. Nevertheless, genuine correlations may exist, undetected by the current study, for example because our relatively young and highly educated participant sample may not be representative enough of the general population. While uncovering such potential relationships would be relevant for our understanding of the links between striatal dopamine and trait characteristics, it seems unlikely those correlations would be strong enough to validate use of the trait measures as approximations of striatal dopamine function if they were not detected with the current participant sample.”

Finally, we have included specific previous findings on the test-retest reliability of FDOPA in the methods section (and also referenced it in the discussion):

“It is a stable measure with good test-retest reliability (intraclass correlation coefficients range from about 0.7 to 0.94 in the striatum) even after a 2-year time interval between acquisitions of scans (Egerton et al., 2010; Vingerhoets et al., 1994).”